# QUEST: Quality-Aware Metropolis-Hastings Sampling for Machine Translation

**Gonçalo R. A. Faria**[1]*,**Sweta Agrawal**[2], **António Farinhas**[2,3],
**Ricardo Rei**[4], **José G. C. de Souza**[4], **André F.T. Martins**[2,3,4,5]
[1]University of Washington, [2]Instituto de Telecomunicações,
[3]Instituto Superior Técnico, Universidade de Lisboa, [4]Unbabel, [5]ELLIS Unit Lisbon
gfaria@cs.washington.edu

## Abstract

An important challenge in machine translation is to generate high-quality and diverse translations. Prior work has shown that the estimated likelihood from the MT model correlates poorly with translation quality. In contrast, quality evaluation metrics (such as COMET or BLEURT) exhibit high correlations with human judgments, which has motivated their use as rerankers (such as quality-aware and minimum Bayes risk decoding). However, relying on a single translation with high estimated quality increases the chances of "gaming the metric". In this paper, we address the problem of sampling a *set* of high-quality and diverse translations. We provide a simple and effective way to avoid over-reliance on noisy quality estimates by using them as the energy function of a Gibbs distribution. Instead of looking for a mode in the distribution, we generate multiple samples from high-density areas through the Metropolis-Hastings algorithm, a simple Markov chain Monte Carlo approach. The results show that our proposed method leads to high-quality and diverse outputs across multiple language pairs (ENGLISH↔{GERMAN, RUSSIAN}) with two strong decoder-only LLMs (ALMA-7B, TOWER-7B).

## 1 Introduction

Machine translation (MT) is becoming increasingly more accurate and powerful, as it benefits from the many capabilities and acquired knowledge of large language models (LLMs) (Freitag et al., 2023b; Hendy et al., 2023). However, for many domains and languages the quality of translation is still not satisfactory—for example, hallucinations or critical errors are a serious issue when translating high-risk content, as in medical and legal domains (Khoong et al., 2019; Taira et al., 2021; Shen et al., 2023; Guerreiro et al., 2023b; Sanz-Valdivieso and López-Arroyo, 2023; Grimm et al., 2024). Developing procedures for sampling higher-quality translations is therefore in high demand.

It is known that the output quality of the translations generated by maximizing the model likelihood is limited because of the *inadequacy of the mode*—models tend to produce distributions over outputs that are highly peaked, favoring a single hypothesis (Eikema and Aziz, 2020; Peters and Martins, 2021; Eikema, 2024); and improving search often makes things worse (Koehn and Knowles, 2017; Stahlberg and Byrne, 2019). In general, maximizing model likelihood tends to overlook hypotheses that could be equally valid and more appropriate in certain contexts. To address this limitation, a string of work has been initiated towards "quality-aware decoding", which leverages powerful quality estimation (QE) and evaluation metrics, such as COMET (Rei et al., 2022) or BLEURT (Yan et al., 2023), to explore and rerank a broader range of candidate hypotheses generated via sampling from the model's distribution, selecting the best-1 (Freitag et al., 2022a; Fernandes et al., 2022b; Farinhas et al., 2023) or the best-$k$ (Jinnai et al., 2024; Singhal et al., 2023) according to these learned metrics.

---

*Work done while at Instituto de Telecomunicações.

38th Conference on Neural Information Processing Systems (NeurIPS 2024).

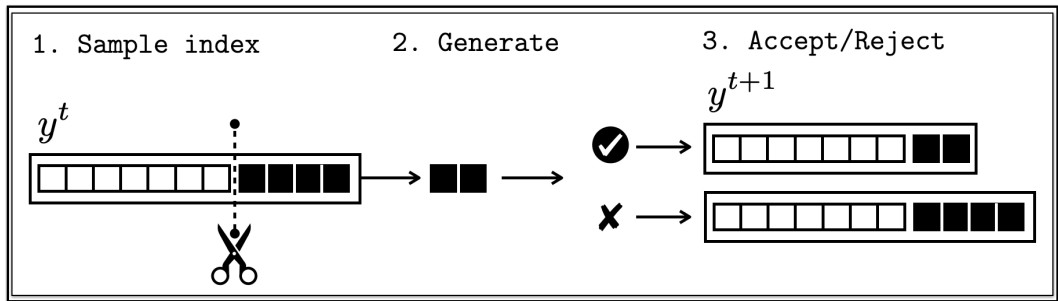

Figure 1: QUEST samples an index from the current translation ($y^t$), removes all elements to the right of the index, generates a new continuation, and then uses the Metropolis-Hastings acceptance criterion to decide whether to accept or reject the resulting new translation. The process continues for a fixed number of $T$ iterations.

Despite the benefits, reranking-based methods have their own drawbacks. There is a risk of these approaches overfitting to the metrics used, potentially leading to illusory gains in quality, as the translations obtained via optimizing these metrics may not always be preferred by humans when compared to other alternatives (Fernandes et al., 2022b). Secondly, their effectiveness is limited by the quality of the initial candidate set. For instance, Vernikos and Popescu-Belis (2024) show that many high-quality translations, generated by combining translation hypotheses, are less likely to be sampled from the model's distribution even with a very large pool size.

One potential way to remedy these issues, which we explore in this paper, is to use these automatic metric proxies as the **energy function of a Gibbs distribution**. However, sampling hypotheses from this distribution presents a difficult challenge: unlike likelihood-based sampling, "quality-based sampling" from a Gibbs distribution cannot be performed autoregressively, and it is intractable to enumerate and score all possible hypotheses. We address this challenge by proposing a simple and effective Markov chain Monte Carlo approach (MCMC) method, combining the Metropolis-Hastings algorithm with a suitable proposal distribution. Our method, **Quality-Aware Metropolis-Hastings (QUEST) Sampling**, uses a novel proposal distribution that is compatible with sentence-level evaluation metrics, common in text generation tasks like MT (Figure 1). We further note that while we focus on MT, where automatic QE metrics are more developed and robust, our proposed method is general and can be applied to other natural language processing (NLP) tasks. Furthermore, as our method is agnostic to the specific quality metric used in the Gibbs distribution, it can directly benefit from any future improvements in the metrics.

We show that QUEST sampling results in high-quality and diverse samples on multiple test beds (WMT23 ENGLISH ↔ {GERMAN, RUSSIAN}) and with multiple decoder-only LLMs (TOWER-7B, ALMA-7B). Our method generates many novel hypotheses from the high-density regions unlikely to be generated via ancestral sampling. Furthermore, with increasing chain size, average quality as measured by automatic metrics continues to improve, unlike ancestral sampling, where the candidate set quality remains unchanged even with a larger pool size.[2]

## 2 Background

### 2.1 Large Language Models for Machine Translation

Generating translations from autoregressive LLMs (either encoder-decoder or decoder-only) involves conditioning the language model on a prompt $x$, which is a sequence of tokens $(x_1, x_2, \ldots, x_L) \in \mathcal{X}$ encoding the text to be translated coupled with a translation instruction (Raffel et al., 2020; Hendy et al., 2023). Let $\mathcal{V}$ be a fixed vocabulary and $\mathcal{Y} = \mathcal{V}^*$ its Kleene closure. The joint distribution over the output translations, $y \in \mathcal{Y}$, given the prompt $x$, can be factorized as the product of conditional probabilities over individual tokens $(y_1, y_2, \ldots, y_N)$, where each $y_i \in \mathcal{V}$. The probability of a

---

[2]We release the code to replicate our experiments at `https://www.questdecoding.com`.

particular translation $y$ for a given input $x$ can be written as

$$p_{\text{LM}}(y|x) = \prod_{i=1}^{N} p_{\text{LM}}(y_i|y_{<i}, x). \tag{1}$$

Here, $y_{<i} := (y_1, y_2, \ldots, y_{i-1})$. During inference, a translation is generated by sampling one token at a time from the distribution $p_{\text{LM}}(y_i|y_{<i}, x)$, adjusted by a temperature, $\tau$. The (adjusted) probability of generating a particular token $y_i$, given the preceding tokens $y_{<i}$ and the input $x$, is defined as:

$$p_{\text{LM},\tau}(y_i = v|y_{<i}, x) = \frac{p_{\text{LM}}(y_i = v|y_{<i}, x)^{1/\tau}}{\sum_{v' \in \mathcal{V}} p_{\text{LM}}(y_i = v'|y_{<i}, x)^{1/\tau}}. \tag{2}$$

Lower temperature values $\tau$ make the distribution more deterministic, favoring the most probable tokens, whereas higher values make the distribution flatter, approximating a uniform distribution.

However, multiple works have scrutinized the reliability of model likelihood as a measure for translation quality (Ott et al., 2018; Stahlberg and Byrne, 2019; Eikema and Aziz, 2020; Freitag et al., 2022a; Eikema, 2024). Instead of solely relying on the likelihood, these works advocate using an automatic translation quality metric as a utility function for minimum Bayes risk (MBR) decoding or reranking based on QE metrics. This shift not only improves the selection of translations but also facilitates the exploration of the underlying distribution learned by the models. However, it is crucial to acknowledge that the overall translation quality is still contingent on the quality of the candidate pool. We next discuss common automatic metrics used for assessing translation quality.

## 2.2 Automatic Metrics for Machine Translation

Automatic quality assessment of machine-generated translations has received considerable attention recently, resulting in metrics that attain high correlations with human judgment of translation quality (Freitag et al., 2022b, 2023b). These automatic metrics are meant to assess the quality of a translation across multiple dimensions (*e.g.* fluency, adequacy) and can provide reliable feedback when human judgments are unavailable. Among these metrics, neural learned metrics that are trained on human translation quality assessment scores or error span annotations have gained significant traction (Rei et al., 2022; Yan et al., 2023; Guerreiro et al., 2023a; Perrella et al., 2022).

For aligning automatically generated translations with human quality preferences, many works have proposed to use automatic metrics as an alternative to human feedback during MT training (Shen et al., 2016; Wieting et al., 2019; He et al., 2024; Xu et al., 2024b) or decoding Shen et al. (2004); Fernandes et al. (2022b); Freitag et al. (2022a, 2023a). Freitag et al. (2022a) show the efficacy of using reference-based neural metrics as utility functions for MBR decoding over lexical alternatives. Further, Fernandes et al. (2022a) use QE metrics like COMET-QE to select 1-best or $N$-best translations from a pool of candidate hypotheses. Their analysis shows that while these automatic metrics can be useful, they might not always reflect human preferences accurately. Additionally, extra care has to be taken when optimizing systems for these metrics, as the improvements might be attributable to overfitting or "gaming the metric", rather than being genuine improvements in translation quality. Nevertheless, these metrics encode useful information about the quality of translations and can still be useful to obtain high-quality translations.

## 3 An MCMC-based Decoding Approach for Text Generation

Given that we have access to an automatic metric that quantifies how desirable a particular translation is, we aim to address the following questions:

*Can we sample directly in proportion to their corresponding quality values? Can we use automatic metrics that already give us a reliable estimate of human-perceived quality to achieve that? Finally, can we use this process to obtain diverse high-quality samples?*

To answer these questions and to address the limitations of quality-aware decoders, we propose to take an alternative approach, quality-aware *sampling*, which ensures high quality and diversity. Recognizing that naturally occurring texts can have numerous valid translations (Nida, 1964; Dreyer and Marcu, 2012; Ott et al., 2018; Mayhew et al., 2020), the ability to generate diverse translation hypotheses is paramount.

To this end, we first present background on Metropolis-Hastings in Section 3.1. Section 3.2 discusses our proposal distribution. Finally, we connect our approach to reinforcement learning with human feedback (RLHF) in Section 3.3.

## 3.1 Metropolis-Hastings

The problem of sampling translation hypotheses in proportion to a metric, $r(x, y)$, can be framed as sampling from the following Gibbs distribution:

$$\pi_\beta(y|x) = \frac{1}{Z_\beta(x)} \exp\left(\frac{r(y,x)}{\beta}\right),$$ (3)

where $Z_\beta(x) = \sum_y \exp\left(\frac{r(y,x)}{\beta}\right)$ and $\beta$ is the temperature of the Gibbs distribution. We denote the corresponding unnormalized density by $\tilde{\pi}_\beta(y|x)$, which unlike $Z_\beta(x)$ can be evaluated for any $(x, y)$.

While several algorithms have been proposed to sample approximately from such intractable distributions (Miao et al., 2018; Berglund et al., 2015; Amini et al., 2023), we resort to Metropolis-Hastings (Metropolis et al., 1953, MH) due to its simplicity and flexibility in handling a wide range of proposal distributions. The MH algorithm generates a sequence of samples from a **target distribution**, here $\pi_\beta(y|x)$, by constructing a Markov chain $(y^0, y^1, \ldots, y^T)$ that has $\pi_\beta(y|x)$ as its equilibrium distribution. It starts from an arbitrary hypothesis $y^0$. In the $t^{\text{th}}$ iteration, it draws a new hypothesis $y$ from a **proposal distribution** $q(y|y^t, x)$. This hypothesis is accepted with an acceptance probability $\alpha_\beta(y, y^t) \in [0, 1]$ given by:

$$\alpha_\beta(y, y^t) = \min\left\{1, \ \frac{\pi_\beta(y|x)\, q(y^t|y, x)}{\pi_\beta(y^t|x)\, q(y|y^t, x)}\right\}.$$ (4)

If the candidate $y$ is accepted, the next state in the chain becomes $y^{t+1} = y$; if rejected, the chain stays at $y^{t+1} = y^t$. The process repeats for some number of steps $T$ and in the end it returns the hypothesis $y^T$. Note that, while computing the likelihood $\pi_\beta(y|x)$ of a particular hypothesis $y$ under the Gibbs distribution is intractable (due to the intractable partition function $Z_\beta(x)$), evaluating the acceptance criterion $\alpha_\beta(y, y^t)$ is easy, because it depends only on the likelihood ratio, in which the normalization constants cancel out, *i.e.*:

$$\frac{\pi_\beta(y|x)}{\pi_\beta(y^t|x)} = \frac{\tilde{\pi}_\beta(y|x)}{\tilde{\pi}_\beta(y^t|x)} = \exp\left(\frac{r(y,x) - r(y^t, x)}{\beta}\right).$$ (5)

MH converges to the unique stationary distribution $\pi_\beta(y|x)$, regardless of the initial distribution we start with, if the transition distribution of the Markov chain, *i.e.*, $p(y^t|y^{t-1}, x) = q(y^t|y^{t-1}, x)\alpha(y^t, y^{t-1})$ satisfies the *Markov chain ergodic theorem* (Neal, 2011).

This requires a suitable proposal distribution $q(y|y^t, x)$ which must be **irreducible** and **aperiodic**. To ensure irreducibility, the proposal distribution should have sufficient support, such that it is possible to transition from any state to any other state with a nonzero probability in a finite number of steps. Aperiodicity ensures that the Markov chain does not get stuck in a cyclic behavior, where it keeps returning to the same states in a fixed pattern. Additionally, due to the acceptance criterion defined in Eq. 4, the transition distribution satisfies the detailed balance condition:

$$\pi_\beta(y|x)\, p(y^t|y, x) = \pi_\beta(y^t|x)\, p(y|y^t, x).$$ (6)

It is trivial to show that the chain has the target distribution, $\pi_\beta(y|x)$ as its stationary distribution:

$$\pi_\beta(y|x) = \sum_{y' \in \mathcal{Y}} p(y|y', x)\pi_\beta(y'|x).$$ (7)

Note that MH can work with almost any proposal distribution. However, if the detailed balance conditions are far from holding, the acceptance probability will be low for transitions, making the convergence to the target distribution very slow and the approach impractical. Hence, choosing a suitable proposal distribution for the task is essential. We next describe our proposal distribution, $q(y|y^t, x)$, which overcomes these limitations and satisfies the required constraints.

## 3.2 Proposal distribution

Previous works (Berglund et al., 2015; Miao et al., 2018; Su et al., 2018) use proposal distributions that generate hypotheses with one-word or token-level modifications. The different formulations in their most basic form propose a Markov chain in which the state comprises the sentence $y^t \in \mathcal{Y}$ and an index variable $i^t \in \{0, \ldots, |y^{t-1}|\}$. At every step, an index is sampled that determines the token to be changed, and a new token is then sampled based on the following full conditional distribution:

$$q(y_i|y_{<i}, y_{>i}) = \frac{\tilde{\pi}(y_{<i}, y_i, y_{>i})}{\sum_{y'_i \in \mathcal{V}} \tilde{\pi}(y_{<i}, y'_i, y_{>i})}, \tag{8}$$

where $\mathcal{V}$ is created by considering the $k$ most likely tokens ($k$ is generally large) under $p_{\text{LM}}(y_i|y_{<i})$.

This procedure, however, has several limitations: first, it makes it difficult to handle more general combinatorial constraints, in our case more sophisticated metrics. As we only explore adjacent positions in the sentence space due to small local changes, the Markov chain risks becoming trapped in infeasible states, necessitating a very large number of steps $T$ to converge. Second, relying solely on token-level modifications makes it exceedingly difficult to generate plausible text, making it impractical to align generations with QE metrics or general reward models. We show that this is indeed the case in Section 5.2.

We instead propose a simple and effective procedure that only requires generating a single hypothesis from the model $p_{\text{LM}}$ and a single evaluation from the metric, $r(y, x)$, and still allows the Markov chain to converge to the target distribution. We characterize the proposal by the following procedure:

1. Given an instance $y^t$ with length $n^t := |y^t|$, sample an index $i$, $i \sim q(i|n^t)$.
2. Generate a completion $y_{\geq i}$ from $p_{\text{LM}}(y_{\geq i}|y^t_{<i}, x)$.

Note that, due to the nature of our proposal distribution, $y^t$ and $y$ share a common substructure (prefix) before the index $i$, i.e., $y^t_{<i} = y_{<i}$, which implies that

$$q(y|y^t, x, i) = p_{\text{LM}}(y_{\geq i}|y^t_{<i}, x) \prod_{j<i} \delta(y_j, y^t_j), \tag{9}$$

where $\delta(y_j, y^t_j)$ is the Kronecker delta function, which assigns zero probability to prefix tokens which are different from $y^t_{<i}$ and probability one to tokens matching the prefix.

The complete proposal when we include the index distribution $q(i|n^t)$, which depends on the previous sentence lengths $n^t := |y^t|$, is given as

$$q(y, i|y^t, x) = q(i|n^t)p_{\text{LM}}(y_{i\geq}|y^t_{<i}, x) \prod_{j<i} \delta(y_j, y^t_j). \tag{10}$$

We will use the uniform distribution as the index distribution, i.e., $q(i|n^t) = 1/n^t$ for each $i \in [n^t]$, unless otherwise stated. Algorithm 1 describes the complete sampling process.

This proposal is both **irreducible** and **aperiodic** as there is a non-zero probability of going from a particular sentence to any other sentence and back. When $i = 0$, we can generate any text from the language model, which, when using ancestral sampling, implies that we can sample any possible sequence. As our approach only requires access to the token-level log probabilities and the ability to generate *good* continuations given a prefix, it can also be used with closed LLMs through an API as long as the it provides access to the sample likelihoods. However, we limit our experiments to open-source models due to the incurred cost and lack of training details about these black-box models.

---

**Algorithm 1** Quality-Aware Metropolis-Hastings (QUEST) Sampling

1: **Input:** $x, p_{\text{LM}}, r, T$
2: **Hyperparameters:** $\tau, t_{\text{burning}}, \beta$
3: Sample initial response $y_0 \sim p_{\text{LM}}(\cdot \mid x)$
4: $t \leftarrow 1$
5: **for** 1 to $T$ **do**
6:      Sample index $i \sim q(i|n^{t-1})$
7:      Sample $y_{\geq i} \sim p_{\text{LM}}(\cdot|y^{t-1}_{<i}, x)$
8:      $y \leftarrow (y^{t-1}_{<i}, y_{\geq i})$
9:      Compute $\alpha_\beta(y, y^{t-1})$ through Eq. 4
10:      Sample $\alpha$ uniformly in $[0, 1]$
11:      **if** $\alpha \leq \alpha_\beta(y, y^{t-1})$ **then**
12:          $y^t \leftarrow y$
13:          $t \leftarrow t + 1$
14:      **end if**
15: **end for**
16: **return** $y^{t_{\text{burning}}}, \ldots, y^t$

---

### 3.3 Connections to Reinforcement Learning with Human Feedback

Reinforcement Learning with Human Feedback (RLHF) leverages human feedback to guide the learning process of complex NLP tasks (Stiennon et al., 2022; Fernandes et al., 2023; Kaufmann et al., 2023). The process is as follows. Given a language model, one generates hypotheses $y \in \mathcal{Y}$ given an input prompt $x$ and gathers human feedback about which outputs are preferable. This preference data is then used to train a proxy reward model $r_\phi(y, x)$. Finally, reinforcement learning (RL) methods are used to optimize the original LM with respect to the reward model, following

$$\max_\pi \mathbb{E}_{x \sim \mathcal{D}, y \sim \pi(y|x)} \left[ \frac{r_\phi(x, y)}{\beta} \right] - D_{\mathrm{KL}} \left[ \pi(y|x) \parallel p_{\mathrm{LM}}(y|x) \right]. \tag{11}$$

Many works show that the optimal solution to the KL-constrained reward maximization objective takes the form (Peters and Schaal, 2007; Peng et al., 2019; Korbak et al., 2022b,a; Go et al., 2023):

$$\pi(y|x) = \frac{1}{Z_\beta(x)} p_{\mathrm{LM}}(y|x) \exp \left( \frac{r(y, x)}{\beta} \right), \tag{12}$$

where $Z_\beta(x) = \sum_{y \in \mathcal{Y}} p_{\mathrm{LM}}(y|x) \exp \left( \frac{r(x,y)}{\beta} \right)$. While we cannot sample autoregressively from this distribution, this density can be represented as a Gibbs distribution with the corresponding reward function $\tilde{r}(x, y) = \log p_{\mathrm{LM}}(y|x) + \frac{r(x,y)}{\beta}$.

Instead of the target distribution expressed in Eq. 3, we could define the above distribution as our target when using QUEST to sample from it, avoiding optimizing the objective in Eq. 11 directly. If we formulate the acceptance criterion using this target density and our proposal distribution introduced in Eq. 10, we obtain the following:

$$\alpha_\beta(y, y^t) = \min \left\{ 1, \exp \left( \frac{r(y, x) - r(y^t, x)}{\beta} \right) \frac{q(i|n)}{q(i|n^t)} \right\}. \tag{13}$$

The full derivation is in Appendix A. Using this approach, we can align language model generations without access to the model weights, log probabilities, or RL. We provide a preliminary experimental comparison using the two target distributions (Eq. 3 and Eq. 12) using QUEST in Appendix C.4.

## 4 Experimental Settings

**Data and Evaluation**    We test our approach on the WMT23 test sets (Kocmi et al., 2023) covering four language pairs, ENGLISH ↔ {GERMAN, RUSSIAN}.

We evaluate the quality and the diversity of the generated texts as follows. Suppose $\bar{\mathcal{Y}}$ is the set of hypotheses generated for the source text $x$ with reference hypothesis $y^\star$ and $\mathcal{D} = \{(x, y^\star, \bar{\mathcal{Y}})\}$ represents the evaluation set. We compute the mean quality over each set of hypotheses using XCOMET-XL (Guerreiro et al., 2023a) as $\frac{1}{|\mathcal{D}|} \sum_{(x,y^\star,\bar{\mathcal{Y}}) \in \mathcal{D}} \frac{1}{|\bar{\mathcal{Y}}|} \sum_{y \in \bar{\mathcal{Y}}}$ XCOMET-XL$(x, y, y^\star)$. Similarly, we measure the mean diversity using the average pairwise BLEU (Papineni et al., 2002; Shen et al., 2019) as $1 - \frac{1}{|\mathcal{D}|} \sum_{(x,y^\star,\bar{\mathcal{Y}}) \in D} \frac{1}{|\bar{\mathcal{Y}}|(|\bar{\mathcal{Y}}|-1)} \sum_{(y,y') \in \bar{\mathcal{Y}}^2 \text{ s.t. } y \neq y'}$ BLEU$(y, y')$.[3]

**Models**    We use TOWER-7B (Alves et al., 2024, `Unbabel/TowerInstruct-7B-v0.2`) and ALMA-7B (Xu et al., 2024a, `haoranxu/ALMA-7B`), two strong decoder-only MT models based on LLAMA2-7B (Touvron et al., 2023) as these models achieve competitive translation quality with GPT-4 and productized models like Google Translate. Unlike ALMA-7B, TOWER-7B uses bilingual MT data as well as datasets from MT-related tasks during training. Prompts are shown in Appendix B.

**Automatic Metrics for QUEST**    We use COMETKIWI-XL (Rei et al., 2023), a reference-free QE model built on top of XLM-R XL (Goyal et al., 2021) and trained to predict human-rated direct assessments (Graham et al., 2013). This metric showed the highest correlations with human judgments on the QE Shared Task organized by the eighth conference on Machine Translation (WMT 2023) Blain et al. (2023). We transform the normalized scores from the QE model into $z$-scores using a logit transformation with clamping applied to mitigate overflow.

---

[3] `https://github.com/mjpost/sacrebleu/tree/master`

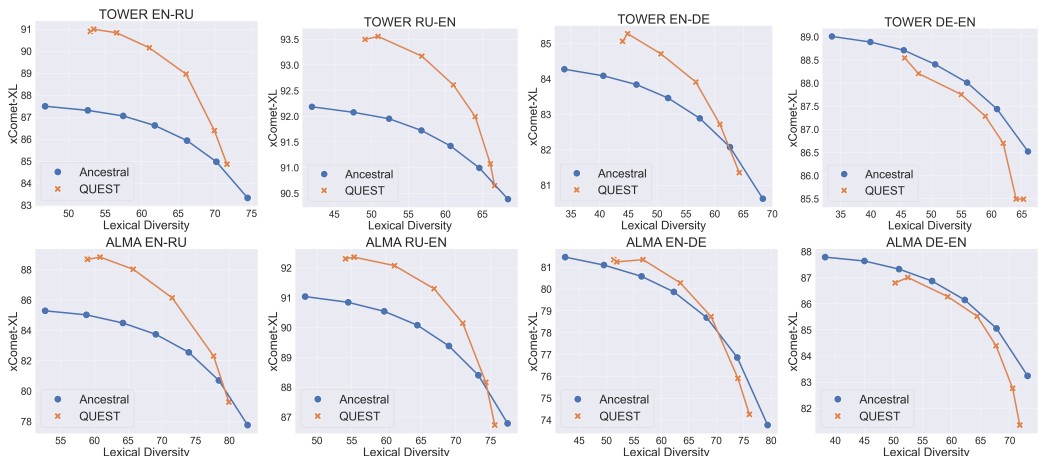

Figure 2: Average quality vs. diversity on WMT23 datasets. Different points represent different hyperparameter values. QUEST outperforms ancestral sampling in six out of eight settings.

**Decoding Configurations** For ancestral sampling, we consider temperature values $\tau$ between $0.2$ and $1.0$, with an equally spaced interval of $0.1$. For generations with QUEST, we sample from the proposal distribution using $\tau = 0.8$ and vary the parameter $\beta$ of the target Gibbs distribution from the following range of values $\{0.01, 0.02, 0.05, 0.1, 0.2, 0.5, 1.0\}$. The number of ancestral samples and decoding steps are both set to 128. We use VLLM (Kwon et al., 2023), a high-throughput and memory-efficient inference engine for generating outputs.

**Compute Comparison: Ancestral Vs QUEST** We use the number of output tokens as a metric to compare the different approaches. For the sake of simplicity, let us assume that the output sentence has a fixed size of $N$. On average, QUEST in $T$ steps generates $\left(\frac{T-1}{2} + 1\right)N = \frac{(T+1)N}{2}$ tokens, $N$ tokens for the initial hypothesis, and then on average $\frac{N}{2}$ tokens for the remaining $T - 1$ steps in the chain. If we contrast against sampling $T$ sentences using ancestral sampling, we decode $T \times N$ tokens. This suggests that for an equal number of samples generated using ancestral sampling and QUEST, the latter results in about $\frac{2T}{(T+1)} \approx 2$ times as many tokens, on average. Note, however, that the computational cost of QUEST is higher than ancestral sampling, as the hypotheses are generated sequentially and evaluated at every step. We run our experiments on NVIDIA RTX A6000 GPUs. Each ancestral sampling and QUEST for run with $T = 128$ takes, on average, 1 hour and 6 hours, respectively, for 2000 unique source texts on a single GPU. The compute bottleneck for QUEST also arises from using a large QE metric, potentially distilling this metric into smaller models could help reduce the compute time. We leave this to future work.

## 5   Results

### 5.1   Main Findings

The main results of our experiments are presented in Figure 2.

QUEST results in better translation quality-diversity trade-offs. Across language directions and models, the samples generated by QUEST tend to have better or similar quality than ancestral sampling as measured by XCOMET-XL.[4] As QUEST does not directly use the reference-based metric, XCOMET-XL, we reduce the chance of overfitting to the metric and thus the gains represent the ability of QUEST to improve translation quality more realistically. We also report Comet-22 vs diversity results in the Appendix Figure 9: the trends remain the same. The benefits of the proposed approach are more noticeable when translating from English (EN → {DE, RU}). Specifically, for EN ↔ RU, our model improves XCOMET-XL by up to 2 points for both language models, showing the efficacy of QUEST over ancestral sampling.

---

[4]We provide results on four additional language pairs in Appendix C.6.

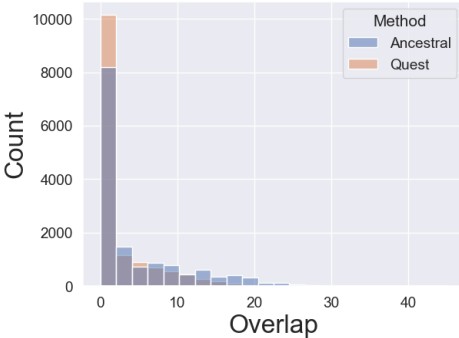

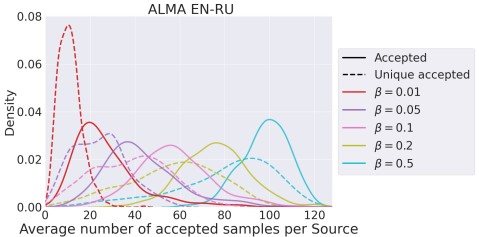

(c) Distribution of # of accepted samples: high $\beta$ results in higher acceptance rates.

(a) Sample Overlap between ancestral and QUEST.

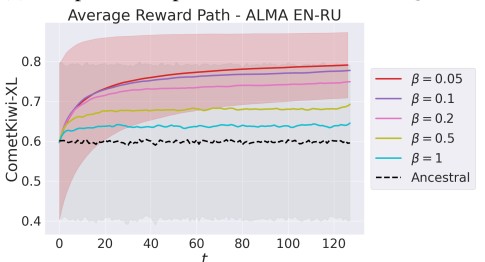

(b) Average quality over MCMC steps and number of ancestral samples $t$.

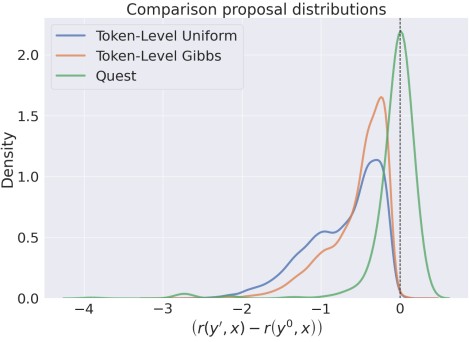

(d) Reward differences between an initial hypothesis and proposal hypotheses given the initial hypothesis.

Although ancestral exhibits better quality than QUEST for DE-EN, further analysis suggests that this discrepancy may stem from the constraints of the QE metric used to model the translation preferences. Notably, QUEST demonstrates significantly higher COMETKIWI-XL scores across the board (see Appendix Figure 7), suggesting that better and more robust QE models can result in improved translation quality.[5] Furthermore, WMT23 DE-EN includes short passages that might require additional steps to reach the high-density regions. Based on a length analysis (See Appendix C.2), we observe that the quality of QUEST lags behind ancestral sampling, specifically for longer sentences. We leave the exploration of finding optimal steps for convergence and adapting the proposal distribution for document-level MT to future work.

## 5.2 Ablation Analysis

We present further analysis on some properties of our proposed approach using WMT23 EN→RU datasets with translations generated using ALMA-7B.

**QUEST generates less-likely high-quality samples.** We compute the set overlap for QUEST and ancestral samples ($T = 128$) with an independent draw of 512 ancestral samples to investigate whether our approach results in hypotheses from unexplored regions (Fig. 3a). Compared to ancestral, QUEST results in hypotheses that are not found in the larger pool (4×) of ancestral samples as illustrated by a higher mass at the overlap value of 0. This demonstrates that QUEST effectively gets to regions of the output manifold that would be less likely sampled by the LM and still attain, on average, better quality and diversity.

**Average reward improves over decoding steps.** Figure 3b shows the average reward with increasing decoding steps and the sample size for QUEST and ancestral respectively. As ancestral results in independent samples, the mean reward estimate remains unchanged for different sample sizes. However, for QUEST, hypotheses are gradually sampled in the direction of higher-quality regions, closer to the target density, resulting in increased average reward with more steps. We can also observe that the standard deviation of the reward over all samples decreases with the increasing number of steps, suggesting that the model eventually reaches a better target distribution.

---

[5] Kocmi et al. (2024) report that a difference of 2.7 COMETKIWI-XL on average is statistically significant with a 98.9% accuracy. QUEST improvements are always greater than this value across the board.

**High $\beta$ value results in higher acceptance rates.** Figure 3c shows the distribution of accepted samples when varying $\beta$, considering all (blue) versus unique (red) accepted samples. As expected, on average, the number of accepted samples is smaller than the number of steps $T$ depending upon the rejection rate—low $\beta$ values lead to higher rejection rates. Furthermore, within the accepted samples, we also observe many repeats. This can be attributed to the observation that the language model has a low entropy distribution for continuations when the sample indices lie at the end of sentences. Moreover, for probable sentences, only a few have a high reward, which could result in the Markov chain getting stuck in a particular state. Increasing the temperature of the LM or adjusting the index distribution to have less density in the last few tokens could potentially reduce the number of repeats. We leave this exploration to future work.

**Our sentence-level proposal generates better candidates.** For a random example sampled from the WMT23 dataset, we generate 25k hypotheses using three proposal distributions: a) token-level modification with uniform probability b) token-level modification with full posterior presented in Eq. 8, and c) our sentence-level proposal (Eq. 10) and calculate the reward differences between the original ($y$) and the generated hypothesis ($y'$): $r(y', x) - r(y, x)$. Figure 3d shows its distribution: for proposals (a) and (b), the reward for the generated hypotheses is almost always lower than the initial translation. On the other hand, our proposal results in assigning half of the probability mass to hypotheses that improve over $y$, leading to faster convergence over token-level alternatives.

## 6 Related Work

**Sampling from Intractable Gibbs distributions.** Several methods have been proposed to sample approximately from Gibbs distributions in text generation using autoregressive and masked-language models. Prior works (Miao et al., 2018; Zhang et al., 2020) use MH with a proposal distribution that makes token-level modifications for constrained generation tasks. Since masked language models (MLM) do not have a straightforward mechanism for sampling text, MCMC has been widely explored using variations of Gibbs sampling (Berglund et al., 2015; Su et al., 2018; Wang and Cho, 2019; Yamakoshi et al., 2022). However, Goyal et al. (2022) shows that the masked conditional distributions from MLMs result in invalid Gibbs samplers and, therefore, proposes to use MH on the masked conditionals, resulting in higher-quality text. Mireshghallah et al. (2022); Forristal et al. (2023) build on this work and use MLMs for sampling from Gibbs distributions. Some works (Kumar et al., 2022; Qin et al., 2022; Amini et al., 2023; Du et al., 2023) also adapt the Hamiltonian MCMC algorithms originally designed for high-dimensional continuous distributions for the discrete scenario (Duane et al., 1987; Neal, 2011). Furthermore, Hu et al. (2024) apply GFlowNets (Bengio et al., 2021) to fine-tune language models for solving posterior inference problems, which can be considered as sampling in proportion to an intractable Gibbs distribution. In our work, we instead aim to use MCMC for MT to sample translations in proportion to a sentence-level evaluation metric.

**Diverse Decoding for Machine Translation.** Variants of beam search (Cho, 2016; Vijayakumar et al., 2017; Kulikov et al., 2019; Tam, 2020) have been proposed to produce a diverse set of translations using diversity-promoting objectives. However, the increased computation cost with the model size and beam width makes it infeasible and impractical to use with LLMs and it fails to yield consistent improvement over ancestral with an increase in beam width (Stahlberg and Byrne, 2019; Eikema and Aziz, 2020; Pang et al., 2024). Quality-aware decoding approaches on the other hand are almost always used to generate a single best hypothesis. Concurrently, Jinnai et al. (2024) add diversity promoting objective to MBR decoding to generate a set of high-quality diverse candidates. However, their approach only allows for the selection of hypotheses from a predefined set. We further note that we do not *directly* promote diversity—rather, diverse translations are a side product of efficiently exploring multiple high-quality regions from the model's distribution.

## 7 Conclusion

We propose a new decoding approach for MT, *Quality-Aware Metropolis-Hastings* (QUEST) sampling based on MCMC that enables the generation of hypotheses in proportion to an automatic QE metric. We present a simple and novel proposal distribution that satisfies the constraints imposed by the Metropolis-Hastings algorithm. Our experiments on four language directions and two strong decoder-only language models show the efficacy of our approach over baselines. We further show that our

approach results in samples from underexplored high-density regions and that the average quality continues to improve as the Markov chain size increases.

# 8 Limitations and Broad Impact

QUEST requires generating many samples to reach high-density regions sequentially from an LLM for each input prompt, which can be computationally expensive for time-critical applications. Additionally, the required number of steps may vary depending on the input and the quality of the initial hypothesis. Furthermore, our proposal distribution only modifies the sentence suffix, which becomes restrictive once the chain reaches a high-quality region. As at this point, only minor changes to the hypothesis are accepted, slowing the mixing process. Addressing these limitations and extending this approach to other NLP tasks are avenues for future work.

Furthermore, we leverage recent advances in QE methods for MT and integrate them directly in the generation process of LLMs, which can potentially reduce the errors generated by these systems. However, despite the high correlations of evaluation metrics with human judgments, they are sometimes hard to interpret and occasionally fail to detect simple mistakes such as incorrect translations of numbers or entities (Amrhein and Sennrich, 2022). In such cases, sampling from the Gibbs distributions induced by these metrics might increase the chances of sampling those erroneous translations. We believe these risks will be mitigated as better metrics are constantly being developed—our method, being agnostic to the specific quality metric, will directly benefit from it. In addition, since QUEST supports any Gibbs distribution, it can also incorporate multiple QE models or additional checks which can rule out problematic samples by assigning them a very low QE score.

## Acknowledgments

We thank Ben Peters, Marcos Treviso, and Sergey Troshin for their helpful and constructive feedback on the initial versions of the paper. Part of this work was supported by the EU's Horizon Europe Research and Innovation Actions (UTTER, contract 101070631), by the project DECOLLAGE (ERC-2022-CoG 101088763), by the Portuguese Recovery and Resilience Plan through project C645008882- 00000055 (Center for Responsible AI), and by Fundação para a Ciência e Tecnologia through contract UIDB/50008/2020.

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

# A   Connection to RLHF derivation

If we take the target distribution to have the following form:

$$\pi(y|x) = \frac{1}{Z(x)} p_{LM}(y|x) \exp\left(\frac{r(x,y)}{\beta}\right), \tag{14}$$

then the likelihood ratio becomes :

$$\frac{\pi(y|x)}{\pi(y^t|x)} = \exp\left(\frac{r(x,y) - r(x,y^t)}{\beta}\right) \frac{p_{LM}(y_{\geq i}|y^t_{<i}, x)}{p_{LM}(y^t_{\geq i}|y^t_{<i}, x)}. \tag{15}$$

Note that the target log ratio contains the inverse of the probability of returning, *i.e.*:

$$\frac{q(y^t|y, x, i)}{q(y|y^t, x, i)} = \frac{p_{LM}(y^t_{\geq i}|y^t_{<i}, x)}{p_{LM}(y_{\geq i}|y^t_{<i}, x)}, \tag{16}$$

this allows simplifying the criterion as follows:

$$\alpha_\beta(y, y^t) = \min\left\{1, \exp\left(\frac{r(x,y) - r(x,y^t)}{\beta}\right) \frac{q(i|n)}{q(i|n^t)}\right\}. \tag{17}$$

# B   Prompts used for MT

For both the ALMA-7B and TOWER-7B we adhere to the prompt format recommended for translation in the original papers as shown below:

ALMA-7B:

```
Translate this sentence from {source language} to {target language}.
{source language} Source: {source sentence}.
{target language} Translation:
```

TOWER-7B

```
<|im_start|>user
Translate the following {source_language} source text to {target_language}:
{source_language}: {source_sentence}
{target_language}: <|im_end|>
<|im_start|>assistant
```

# C   Additional Results

## C.1   Toy problem: Validating the Approach

To validate that our approach works, we test QUEST on a controlled setting, a toy summarization problem, where the ground truth reward is known. We consider the following reward function:

$$r(x,y) := \text{logit}\left(\text{clamp}\left(\mathcal{N}\left(|y|; \mu = 7.5, \sigma^2 = 3.75^2\right)\right)\right), \tag{18}$$

where $\text{clamp}(x) = \max(10^{-2}, \min(x, 1 - 10^{-2}))$.

We use GPT2 (Radford et al., 2019) fine-tuned on the Reddit dataset Stiennon et al. (2020) to generate summaries for 128 examples randomly sampled from the test split. We compare the distribution of rewards obtained by different sampling techniques (Ancestral, QUEST and Truncated-Gibbs) to the ground truth reward in Figure 4a. For Truncated-Gibbs sampling, we estimate the partition function using the ancestral samples and resample them based on the probability distribution induced by the rewards. Unlike ancestral sampling which results in samples that are far from the high-reward regions, our sampling strategy, QUEST, results in the best approximation of the ground truth distribution. This simplified reward task serves as an useful example to show the efficacy of our approach over alternatives.

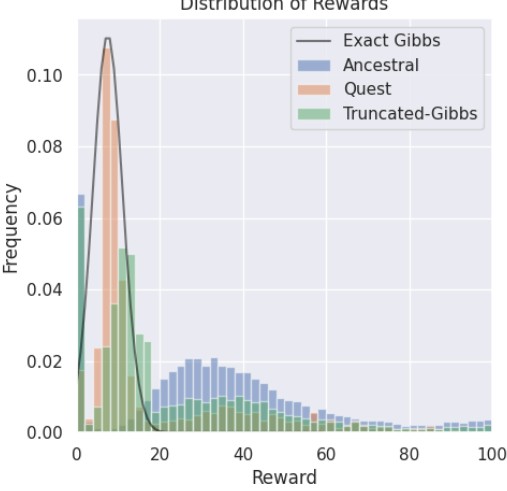

(a) Distribution of rewards for sampled hypotheses for the toy summarization problem.

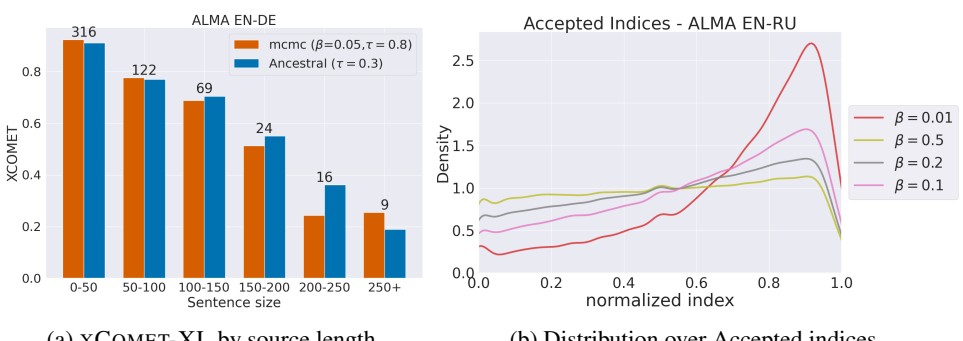

(a) XCOMET-XL by source length.

(b) Distribution over Accepted indices.

## C.2 MCMC results in better outputs for shorter segments.

We show the XCOMET-XL scores on WMT23 English-German pairs bucketed by the length of the source text. In general, the quality of samples degrades with increased source length when decoding via MCMC as shown in Figure 5a. One possible factor could be the uniform index-selection exploration strategy where the proposal might require more steps as the sentence length increases. Another factor could be the limitations of the reward function itself for selecting high-quality translations for longer sequences. As these quality estimation metrics have not been trained on longer sequences, due to a distribution shift, they might not be representative of true quality for these output lengths. We believe the latter option could be the more significant factor as the reward optimized still improves regardless of the sentence length (see Figure 7).

## C.3 Accepted proposals are skewed towards the end of the sentence.

Figure 5b illustrates how the distribution of accepted indices changes with decreasing $\beta$: lower $\beta$ values tighten acceptance criteria, favoring states with higher rewards and a greater likelihood of returning to the previous state. Consequently, transitions tend to yield minor alterations, typically at the end of the sentence. Altering the beginning necessitates sampling small indices values, risking a complete sentence change. This plot highlights the current limitations of our proposal, especially the downstream implications on the diversity of prefixes we get for a particular motivating the use of parallel chains and potential avenues for improvement for future work.

## C.4 Comparing RLHF-QUEST vs QUEST

We compare QUEST with the alternative discussed in Section 3.3 where the target distribution in Eq. 3 is replaced with Eq. 12, referred to as "RLHF-QUEST". We compare sampling using the two target distributions in Figure 6. Our results indicate that both approaches result in comparable translation quality, with QUEST resulting in slightly higher diversity on average than RLHF-QUEST. We leave the detailed exploration of these two methods to future work.

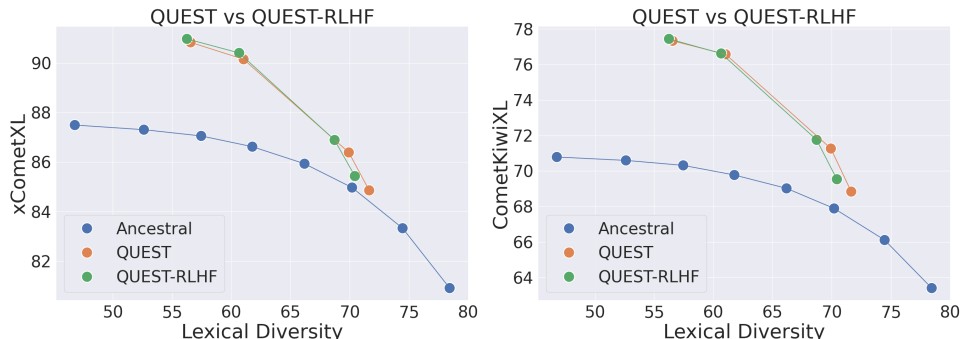

Figure 6: Average Quality by XCOMET-XL (left) and COMETKIWI-XL on English-Russian dataset using TOWER-7B

## C.5 QE Results

We report the quality as measured by the metric used in sampling, *i.e.*, COMETKIWI-XL in Figure 7. QUEST results in higher quality across the board compared to ancestral sampling.

## C.6 Additional Language Pairs

We further expanded our evaluation to four additional language pairs, as shown in Figure 8: WMT23 English-Chinese (EN→ZH, high-resource), WMT23 English-Czech (EN→CS, medium-resource), WMT22 English-Icelandic (EN→IS), and Icelandic-English (IS→EN, low-resource), using ALMA-7B. We did not include TOWER in these experiments because it was trained on the WMT22 test sets. Across all language pairs, QUEST consistently outperforms ancestral sampling, providing better quality-diversity trade-offs.

## C.7 COMET22 Results

We report the quality as measured by COMET22 (Rei et al., 2022) in Figure 9. QUEST results in higher quality across the board compared to ancestral sampling.

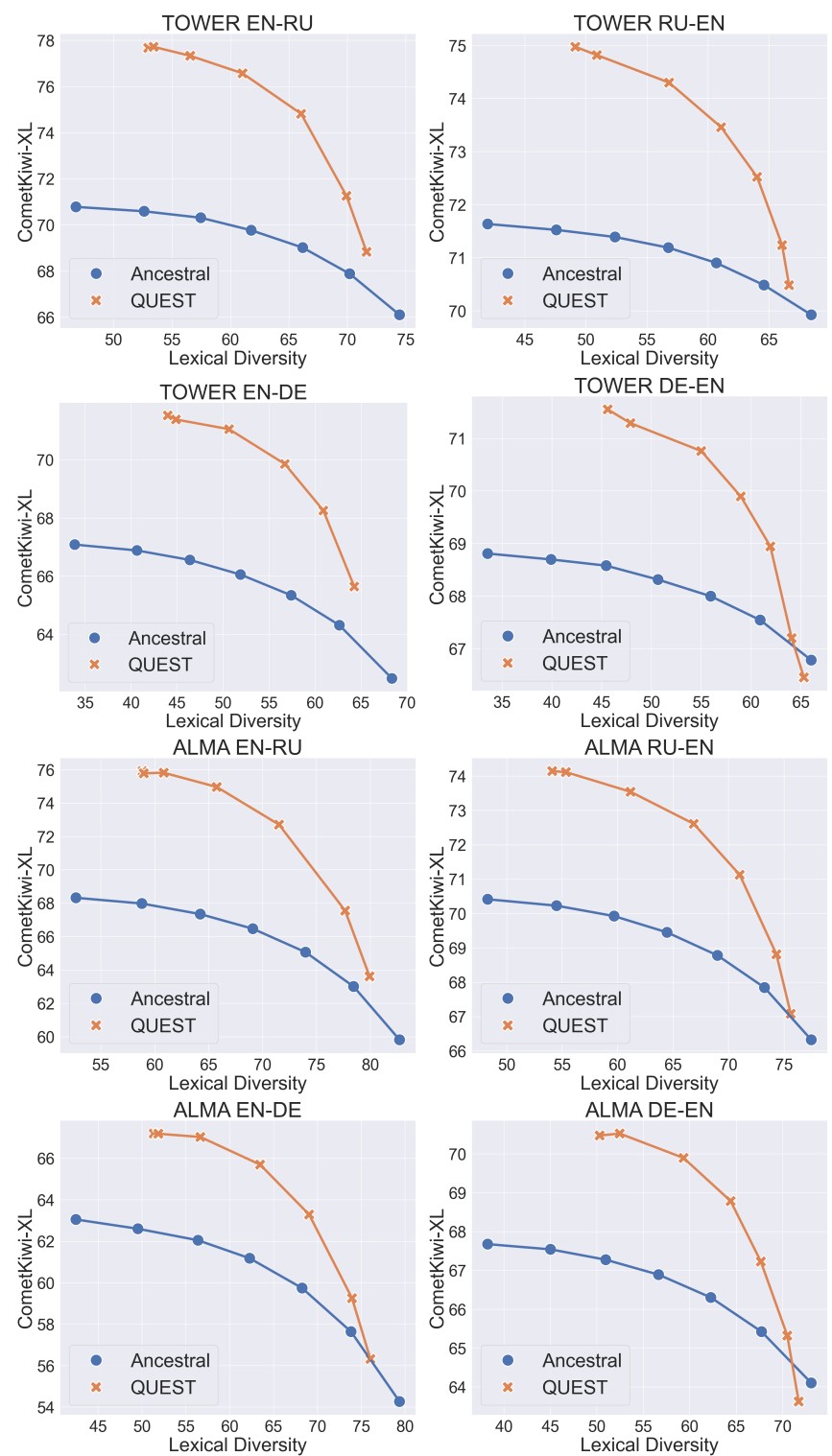

Figure 7: Average quality (COMETKIWI-XL) vs. diversity (PAIRWISE-BLEU) on WMT23 datasets. Different points represent different hyperparameter values.

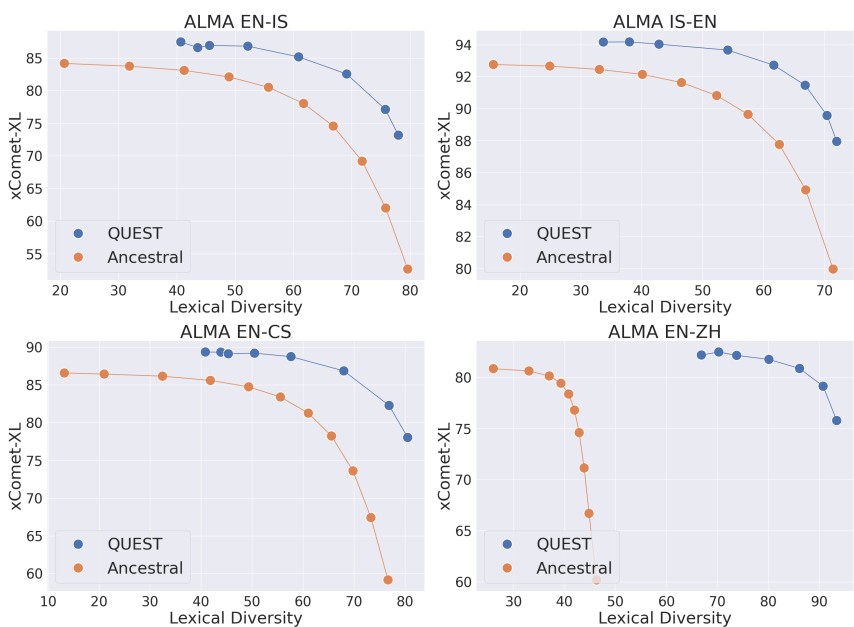

Figure 8: Average quality (XCOMET-XL) vs. diversity (PAIRWISE-BLEU) on additional LPs from WMT23 and WMT22. Different points represent different hyperparameter values.

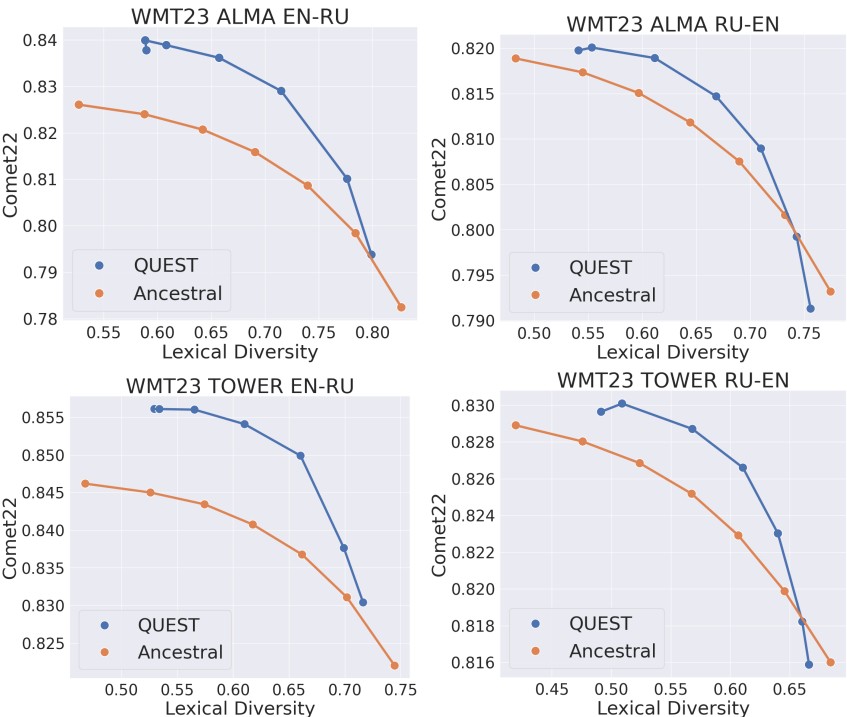

Figure 9: Average quality (COMET22) vs. diversity (PAIRWISE-BLEU) on EN→RU and RU→EN. Different points represent different hyperparameter values.

