# OpenReview forum: "QUEST: Quality-Aware Metropolis-Hastings Sampling for Machine Translation"
_NeurIPS.cc/2024/Conference — NeurIPS 2024 poster_

### Official Review · Reviewer_SYtr · 2024-07-12

**Soundness:** 3
**Presentation:** 3
**Contribution:** 3
**Rating:** 7
**Confidence:** 3

**Summary:**

This paper proposes novel quality-aware sampling for neural machine translation, namely Quality-Aware Metropolis-Hastings Sampling. The main idea is that sampling from a model in proportion to a metric can be seen as sampling from a Gibbs distribution, and Metropolis-Hastings MCMC algorithm can be used for this purpose. Empirical results on 4 WMT language pairs show that QUEST can generate diverse high-quality set of hypothesis compared to the ancestral sampling.

**Strengths:**

- Paper is well-written and easy to follow
- Novel idea for sampling a set of high-quality, diverse hypothesis rather than re-ranking using quality metrics
- Comparison with the ancestral sampling shows the usefulness of the proposed method

**Weaknesses:**

- While you have a comparison on the sampling quality/diversity, I would also be interested in the performance of the translation model on common translation metrics.

(Please note here that I consider all weaknesses to be minor)

**Questions:**

- While average quality and diversity can be higher than for ancestral sampling, does it result in the general higher quality of the MT?
- Figure 1: `Different points represent different hyperparameter values`: I am not sure I understand to what hyperparameters this refers to. Are they the same across al plots?
- Line 247-248: `Note, however, that the computational cost of QUEST is higher than ancestral sampling` - do you know the rough speed ratio between those two?

**Limitations:**

N/A, Limitations are addressed in section 8

---

> ### Author Rebuttal · Authors · 2024-08-07
>
> Thank you for your positive and insightful comments. We address your questions below:
>
>  > “While you have a comparison on the sampling quality/diversity, I would also be interested in the performance of the translation model on common translation metrics.”
>
> This is a good suggestion. We report additional machine translation metrics in Figure 2  in the attached pdf (although our choice of xCOMET-based metrics in our submission is justified by the fact that they were found to be higher correlated with human judgments of translation quality, over COMET22 and BLEURT,  in WMT23 Metrics Shared Task Campaign). We observe that the trends are mostly similar to xCOMET-XL, which further validate our claims.
>
>  > “Figure 1: Different points represent different hyperparameter values: I am not sure I understand to what hyperparameters this refers to. Are they the same across al plots?”
>
> For ancestral, these refers to different temperature values and for QUEST, different beta values.
>
>  > “Line 247-248: Note, however, that the computational cost of QUEST is higher than ancestral sampling - do you know the rough speed ratio between those two?”
>
> You are right and we acknowledge this in L249: for 2000 source texts, QUEST can be 6x slower.
>
>  > “While average quality and diversity can be higher than for ancestral sampling, does it result in the general higher quality of the MT?”
>
> We report the best-of-n translations (using reference-based xCOMET-XL) for the 128 candidates generated using ancestral and QUEST in Figure 3 in the attached PDF. The overall quality of translations is better for ancestral and QUEST for EN-ZH and EN-CS, respectively.  Our current results show that the best-of-n according to an automatic metric is not always higher for QUEST.

---

> > ### Comment · Reviewer_SYtr · 2024-08-12
> >
> > Thank you for your reply and providing additional results. After carefully reading rebuttal response and other reviewers comments, I decided to keep my score unchanged

---

### Official Review · Reviewer_DVBo · 2024-07-13

**Soundness:** 3
**Presentation:** 3
**Contribution:** 3
**Rating:** 6
**Confidence:** 3

**Summary:**

This paper proposes a novel approach called Quality-Aware Metropolis-Hastings (QUEST) Sampling, using a proposal distribution that is compatible with sentence-level metrics. The authors conducted experiments in the machine translation task with four directions En <> {Ru, De} and employed multiple decoder-only LLMs (Tower-7B and ALMA-7B). The experimental results show that the proposed approach leads to high-quality and diverse outputs.

This paper is well organized with focused contributions. The proposed approach could be applied to text generation tasks, though experiments are carried out in machine translation tasks. It is unclear how this approach is robust in low-resource directions or longer sequence text generation. Further discussion would be helpful to understand prros/cons of the proposed approach.

**Strengths:**

- well-organized paper with focused contributions
- technically sound in most cases, except De->En direction. Experimental results show the efficiency of QUEST.

**Weaknesses:**

- this is not a major concern though this work is limited to sentence-level metrics. Since LLMs can handle longer text sequences, discussion on how to extend this idea to document-level metrics would be interesting.
- This approach might work well with high-resource data. How do you overcome in case of data scarcity?

**Questions:**

- Since LLMs can handle longer text sequences, discussion on how to extend this idea to document-level metrics would be interesting. What kind of challenges would lie? From Section 5.1, QUEST might struggle more as sentence gets longer?
- Do you think if this proposed approach works well with a small amount of data?
- This approach might work well with high-resource data. How do you overcome in case of data scarcity?
- Are those tested translation tasks considered high-resource? How robust does this approach in low-resource directions?

---

> ### Author Rebuttal · Authors · 2024-08-07
>
> > “Since LLMs can handle longer text sequences, discussion on how to extend this idea to document-level metrics would be interesting. What kind of challenges would lie? From Section 5.1, QUEST might struggle more as sentence gets longer?”
>
> This is a very good question. Please note that, in Appendix Section C.3, we analyzed how our method performs on short passages present in WMT23 English-German dataset (corresponding to L269) and hypothesized that QUEST could benefit from more document tailored proposal distributions (e.g. sampling an entire sentence) or a larger number of iterations. We also note the limitation of existing sentence level metrics in providing reliable assessment on such texts. We agree and appreciate the suggestion on using document level metrics (for eg SLiDE [1]) for handling longer text translations in future work. We will also add this discussion in the paper.
>
> [1] SLIDE: Reference-free Evaluation for Machine Translation using a Sliding Document Window (Raunak et al., NAACL 2024).
>
>  > “Do you think if this proposed approach works well with a small amount of data? This approach might work well with high-resource data. How do you overcome in case of data scarcity? Are those tested translation tasks considered high-resource? How robust does this approach in low-resource directions?”
>
> Our proposed method is an inference-only strategy independent of the quantity of the data for the language pair being evaluated. It only depends upon two factors 1) The underlying LLM should be able to generate a reasonable set of hypotheses (good and bad quality) and 2) the reward metric should be able to provide a reliable indication of quality.  Recent work has shown that LLMs like ALMA are capable of generating reasonable quality translations for low-resource language pairs with small number of parallel data (~10k) albeit not to the extent of higher-resource language pairs like English-German or English-Russian.
>
> To address your question, we ran new experiments in four more language pairs, including a low-resource setting of English-Icelandic and Icelandic-English. We evaluate our method against ancestral with CometKiwi-XL. The results are shown in Figure 1 in the attached pdf. We find that QUEST can provide a slightly better quality-diversity tradeoffs than ancestral even in this low-resource scenario. However, this does not happen for all values of beta for QUEST. We hypothesize that this could be because COMETKiwi-XL might not be a good reward model for this setting. While the underlying pretrained model that supports CometKiwi-XL is trained on Icelandic, CometKiwi-XL is not trained on any human assessment for English-Icelandic language pair which might impact the quality of judgments generated for this language pair for diverse translations. So, a reward model trained on human assessments for the specific language pair should yield better results.

---

### Official Review · Reviewer_Pz9P · 2024-07-13

**Soundness:** 3
**Presentation:** 3
**Contribution:** 3
**Rating:** 6
**Confidence:** 3

**Summary:**

This paper presents a novel approach called QUEST Sampling, designed to generate high-quality and diverse translations in machine translation. The authors proposed methods to obtain high-quality and diverse parallel data and provide an effective way to avoid over-reliance on noisy quality estimates by using them as the energy function of a Gibbs distribution.

**Strengths:**

The authors propose QUEST sampling, which addresses a significant issue in machine translation: the bias that arises when reranking by a quality-estimated model. This bias occurs because both the sampling and evaluation processes use the same metric.

**Weaknesses:**

1. The authors may consider expanding their experiments to include a wider range of language pairs beyond just German and Russian.

2. Since the sampling is still based on the quality-estimated model, there remains a risk of "gaming the metric." It would be beneficial for the authors to include human evaluations to validate their results.

**Questions:**

N/A

---

> ### Author Rebuttal · Authors · 2024-08-07
>
> Thank you for the constructive feedback and insightful comments.
>
>  > “The authors may consider expanding their experiments to include a wider range of language pairs beyond just German and Russian.”
>
> We agree that expanding the evaluation to more language pairs will strengthen our paper further.  In the original submisison, we tested on 4 language directions (EN-DE, DE-EN, EN-RU and RU-EN) using two strong decoder-only MT models (ALMA and Tower).  We now followed your suggestion and extended the evaluation to four new language pairs (please see Figure 1 in the attached pdf). This includes English-Chinese (high-resource), English-Czech (medium resource), English-Icelandic and Icelandic-English (low-resource). We observe that QUEST still results in a slightly better quality-diversity tradeoffs compared to ancestral sampling for beta values corresponding to the medium diversity regime. In the very high diversity regime, the results of QUEST degrade -- we posit that this is because CometKiwi-XL cannot provide a reliable indication of translation quality for very diverse samples, as it was not trained to predict human quality assessments for these language pairs. Improving the reward model should further improve the quality of translations generated by our method, as suggested by the experiments in EN-DE, DE-EN, EN-RU and RU-EN. We will add a discussion to comment on these new results.
>
>  > “Since the sampling is still based on the quality-estimated model, there remains a risk of "gaming the metric." It would be beneficial for the authors to include human evaluations to validate their results.”
>
> We emphasize that our method, QUEST, is designed to decrease this risk compared to existing methods in the following ways:
>
>  **1.** Unlike popular reranking approaches, we do not use the metric directly to select the hypothesis, instead we sample from the distribution induced by the reward model.
>
>  **2.** We are not aiming to generate one best hypothesis that might overexploit metric biases but rather a pool of translations which are sampled from the high-quality (according to reward) regions of the underlying model.  By doing this, we try to ensure that the translations are not narrowly tailored to specific metrics but are generally robust and of high quality.
>
> However, we acknowledge that this risk still exists: for example, Figure 4 in the appendix shows that the gains obtained in the metric optimized for (reference-less CometKiwi-XL) are much larger. Yet, these gains are also validated via translation quality improvements when measured in the reference-based evaluation metric, XCOMET-XL. We do agree that human evaluation would provide further validation of our claims. However, these would require annotating large sets of translation hypotheses for each source sentence,  which is extremely costly and could not fit our  budget and time constraints.

---

> > ### Comment · Reviewer_Pz9P · 2024-08-13
> > **Thanks for response**
> >
> > I thank the authors for their response. I keep my positive view and maintain my scores.

---

### Official Review · Reviewer_L9No · 2024-07-13

**Soundness:** 2
**Presentation:** 2
**Contribution:** 2
**Rating:** 5
**Confidence:** 4

**Summary:**

This essay proposes one method to solve the challenge of balancing the generation quality and diversity of machine translation.  This essay proposes this problem of sampling a set of high-quality and diverse translations. It is said that this proposed method can lead to high-quality and diverse outputs.

**Strengths:**

- The methodology and formula of this essay are detailed and makes sense.

**Weaknesses:**

Same with questions.

**Questions:**

-- The motivation of balancing the quality and diversity of machine translation is good but difficult. In fact, the quality of machine translation is good. So, can we induce that your method aims to improve the diversity of mt, which seems not very promising.
-- In your method, you will sample from a set of high-quality and diverse translations. Where do these translations come? What is the retrieval set? Or How to generate them?
-- It will be better if there is one main figure to illustrate your method.
-- The formula is good and makes sense. However, adding more examples will be more clear.
-- The evaluation metrics of QUEST seem not very popular.
--If you can provide more results of other common datasets, it will be convincing.

---

> ### Author Rebuttal · Authors · 2024-08-07
>
> Thank you for your comments.
>
> > “In fact, the quality of machine translation is good. So, can we induce that your method aims to improve the diversity of mt, which seems not very promising.”
>
>
> The goal of our paper is to maintain or improve MT quality while increasing the diversity of the generated translations. We believe that this is an important problem which has been overlooked in the literature. Naturally occurring texts have many possible valid translations (L112-113) and in many practical scenarios, providing alternative translations can improve communication and user experience [1] or can even be combined in parts to potentially generate even better translation hypotheses [2]. Existing methods either sacrifice quality to achieve diversity or aim only at high-quality translations that are not diverse. Our proposed method achieves both.
> We show an example in Figure 1 for a commonly used method, ancestral sampling. As shown in Figure 1, increasing the temperature for ancestral sampling results in diverse but overall lower quality pool of samples. On the other hand, our approach (QUEST) improves both the quality and diversity over ancestral samples in six out of eight settings, by incorporating a reward signal in the generation process.
> We hope this alleviates your concern.
>
>
>  > “In your method, you will sample from a set of high-quality and diverse translations. Where do these translations come? What is the retrieval set? Or How to generate them?”
>
>
> This is explained in  lines 125-132 and lines 165-169 in Section 3, where we give full details about how and where these high quality diverse translations come from. There is no retrieval involved.
> We recap here what our procedure is (also stated as Algorithm 1 in the paper):
>
>  **1.** Sample an initial translation from the model using ancestral sampling (e.g. y_1 = The cat sits on the mat)
>
>  **2.** Select a position in the candidate (e.g. 4)
>
>  **3.** Regenerate continuation from the LLM starting that position (e.g. y_2 = The cat sits **behind the sofa**)
>
>  **4.** Score both y_1 and y_2 using an automatic metric (s_1 and s_2) and extract their likelihoods (l_1 and l_2).
>
>  **5.** Compute the ratio in  equation 4; if alpha > some threshold, the new hypothesis is y_2 else it is y_1.
>
>  **6.** Repeat steps 2-5 for k iterations.
>
> At the end of k iterations, we get x (<k) number of high-quality diverse translations.
> We hope this clarifies.
>
>  > “It will be better if there is one main figure to illustrate your method.”
>
> This is a good suggestion. We will add a figure in the paper illustrating the procedure above, to provide more intuition about how our method works.
>
>  > The evaluation metrics of QUEST seem not very popular. If you can provide more results of other common datasets, it will be convincing.
>
>
> We respectfully disagree with the claim that the evaluation metrics are not “popular” and the datasets are not “common”. We use the most recent WMT23 datasets for our evaluation, as well as state-of-the-art metrics (xCOMET-XL), which obtained the highest correlations with human judgments in WMT metrics shared task 2023 (see [3]). These datasets and metrics to the best of our knowledge are the most recent and best methods for evaluating machine translation.
> Nevertheless, we also provide COMET22 vs diversity plots in the attached pdf (Figure 2). As you can see, the trends remain the same. We will add these plots to the final version as an appendix.
>
>
> [1] Ge Gao, Bin Xu, David C. Hau, Zheng Yao, Dan Cosley, and Susan R. Fussell. 2015. Two is Better Than One: Improving Multilingual Collaboration by Giving Two Machine Translation Outputs. In Proceedings of the 18th ACM Conference on Computer Supported Cooperative Work &amp; Social Computing (CSCW '15). Association for Computing Machinery, New York, NY, USA, 852–863. https://doi.org/10.1145/2675133.2675197
>
> [2] Giorgos Vernikos and Andrei Popescu-Belis. Don't Rank, Combine! Combining Machine Translation Hypotheses Using Quality Estimation. arXiv preprint arXiv:2401.06688 (2024).
>
> [3] Results of WMT23 Metrics Shared Task: Metrics Might Be Guilty but References Are Not Innocent (Freitag et al., WMT 2023)

---

### Author Rebuttal · Authors · 2024-08-07

We thank all the reviewers for their constructive and helpful feedback. We have attached a pdf to support additional experiments performed during the rebuttal period.

We are glad that the reviewers found our approach to be novel (SYtr), the paper well-written and well-organized (SYtr, DVBo, L9No) and the method technically sound (DVBo).

We believe the revisions will significantly improve the quality and clarity of our paper. We will release the code to facilitate the reproducibility of our results on acceptance. If we have succeeded in responding to your comments, kindly consider raising the scores. We are happy to address any more questions you might have.



Thank you,

Authors

---

### Decision · Program_Chairs · 2024-09-25

**Decision:**

Accept (poster)

**Comment:**

This paper proposes QUEST (Quality-Aware Metropolis-Hastings) Sampling, a novel approach for generating diverse and high-quality machine translations.

It uses evaluation metrics as the energy function of a Gibbs distribution to avoid over-reliance on noisy quality estimates; employs the Metropolis-Hastings algorithm to generate multiple samples from high-density areas; demonstrates quality-diversity trade-offs compared to ancestral sampling across multiple language pairs and models.

**Strengths:** Novel idea for sampling diverse, high-quality hypotheses rather than just reranking; technically sound approach with strong empirical results; addresses an important challenge in MT of balancing quality and diversity.

**Weaknesses:** Limited language pair evaluation in initial submission (addressed in rebuttal); lack of human evaluation to validate metric-based improvements; computational cost; potential struggles with longer sentences and low-resource settings.